# Quantitative Determination of 2-Oxo-Imidazole-Containing Dipeptides by High-Performance Liquid Chromatography/Tandem Mass Spectrometry

**DOI:** 10.3390/antiox11122401

**Published:** 2022-12-02

**Authors:** Somei Komae, Shingo Kasamatsu, Koji Uchida, Hideshi Ihara

**Affiliations:** 1Department of Biological Science, Graduate School of Science, Osaka Prefecture University, Osaka 599-8531, Japan; 2Department of Biological Chemistry, Graduate School of Science, Osaka Metropolitan University, Osaka 599-8531, Japan; 3Graduate School of Agricultural and Life Sciences, The University of Tokyo, Tokyo 113-8657, Japan

**Keywords:** 2-oxo-imidazole-containing dipeptides, imidazole-containing dipeptides, carnosine, anserine, balenine, homocarnosine, homoanserine, meat, mass spectrometry, stable isotope dilution method

## Abstract

2-Oxo-imidazole-containing dipeptides (2-oxo-IDPs), novel imidazole-containing dipeptide (IDP) derivatives, exhibit a much higher antioxidant capacity than that of IDPs. However, quantitative methods have only been developed for IDPs, and methods for the quantitative analysis of 2-oxo-IDPs are needed. In this study, we developed methods for the quantitative analysis of 2-oxo-IDPs by high-performance liquid chromatography with online electrospray ionization-tandem mass spectrometry (HPLC-ESI-MS/MS) coupled with a stable isotope dilution method. First, we prepared stable isotope-labeled IDP and 2-oxo-IDP standards for MS analyses. Next, using these standards, we established highly sensitive, selective, and absolute quantitative analysis methods for five IDPs and five 2-oxo-IDPs by HPLC-ESI-MS/MS, achieving a limit of detection in the fmol range. Finally, we applied the method to various types of meat, such as beef, pork, chicken, and whale meat, demonstrating the detection of both IDPs and 2-oxo-IDPs. Furthermore, we provide the first evidence for the endogenous production of 2-oxo-balenine in meats. The methods developed in this study enable the detection of trace levels of 2-oxo-IDPs in biological samples and could be helpful for understanding the biological relevance of 2-oxo-IDPs.

## 1. Introduction

Imidazole-containing dipeptides (IDPs) are endogenous dipeptides composed of histidine or its methylated derivatives. To date, five types of IDPs have been reported. Carnosine (β-alanyl-*l*-histidine), a representative IDP, was first discovered in beef extract in the 1900s [1], and its methylated analogs, anserine (β-alanyl-3-methyl-*l*-histidine) and balenine (β-alanyl-1-methyl-*l*-histidine) were identified in goose and snake samples [2,3]. Homocarnosine (γ-aminobutyryl-*l*-histidine) and homoanserine (γ-aminobutyryl-3-methyl-*l*-histidine), in which β-alanine is replaced by γ-aminobutyric acid (GABA), were first isolated from bovine brain samples [4,5]. Several IDPs are endogenously produced in vertebrates; they are abundant in the skeletal muscle and brain and have been detected in other tissues, such as the heart, kidneys, and liver [6,7]. The metabolism of IDPs is strictly controlled by specific enzymes, such as carnosine synthase, carnosine *N*-methyltransferase, and carnosinase [8,9,10]. Accumulating evidence has indicated that IDPs are widely distributed and precisely regulated by enzymes, suggesting that they act as important functional molecules in vivo.

Numerous studies have identified the biochemical functions of IDPs, including roles in pH-buffering [11,12], heavy metal chelation [13,14], anti-glycating activity [15,16], and antioxidant activity [17]. IDPs have protective effects against various oxidative stress-related diseases, such as diabetes, ischemia, cancer, aging, and neurodegenerative diseases [6]. Nevertheless, in vitro assays of their antioxidant capacity have yielded inconsistent results, suggesting that IDPs exhibit markedly lower antioxidant activity than that of other endogenous antioxidants, such as glutathione and ascorbic acid [18,19,20,21,22]. This discrepancy is an important unresolved issue.

Recently, we revealed that 2-oxo-imidazole-containing dipeptides (2-oxo-IDPs), such as 2-oxo-carnosine, 2-oxo-anserine, 2-oxo-homocarnosine, and 2-oxo-homoanserine, are endogenously produced in mouse tissues as a novel oxidized derivative of IDPs by high-performance liquid chromatography with online electrospray ionization-tandem mass spectrometry (HPLC-ESI-MS/MS) coupled with a stable isotope dilution method [7,23]. 2-Oxo-carnosine and 2-oxo-anserine have also been detected in several types of meat samples [24]. Notably, multiple in vitro assays have demonstrated that 2-oxo-carnosine and 2-oxo-anserine exhibit a greater antioxidant capacity than that of corresponding IDPs, suggesting that 2-oxo-IDPs play a key role in the antioxidant capacity of IDPs [25]. In addition, 2-oxo-homoanserine also have showed potent radical scavenging capacity than the precursor, homoanserine [23]. These findings emphasize the potentials of the bioactive substances for developing preventive and therapeutic strategies for oxidative stress-related diseases such as Alzheimer’s disease. However, further studies to unveil the physiological functions of 2-oxo-IDPs are currently difficult due to the limited commercially availability of several precursor IDPs (i.e., balenine, homocarnosine, and homoanserine). Thus, the development of a simple and efficient procedure to prepare standards of IDPs and its oxidized derivatives, 2-oxo-IDPs, with high yield and high purity, is required.

Numerous studies have demonstrated the detection and quantification of IDPs using HPLC coupled with various detection systems, such as ultraviolet (UV) detectors [26,27] and fluorescence detectors [28,29]. Furthermore, HPLC coupled with MS/MS has also been applied for the detection and quantitation of IDPs [30]. However, a highly sensitive detection system for 2-oxo-IDPs is lacking. In this study, we developed a sensitive, highly selective, and absolute quantitative analysis method for 2-oxo-IDPs using HPLC-ESI-MS/MS coupled with a stable isotope dilution method. We also synthesized stable isotope-labeled IDPs and 2-oxo-IDPs as standards. Furthermore, using stable isotope labeled-standards and the newly developed method, we quantified the IDP and 2-oxo-IDP contents in several meat samples, such as beef, pork, chicken, and whale meat.

## 2. Materials and Methods

### 2.1. Materials

Fresh beef (loin, Australia), pork (loin, Japan), chicken (breast, Japan), and minke whale (red meat, Southern Ocean) samples were purchased from various suppliers around the world. Di-*tert*-butyl dicarbonate and 1-ethyl-3-(3-dimethylaminopropyl)carbodiimide monohydrochloride (EDC) were obtained from Peptide Institute (Osaka, Japan). β-Alanine, γ-aminobutyric acid, and *l*-ascorbic acid were obtained from Nacalai Tesque (Kyoto, Japan). *N*-*tert*-Butoxycarbonyl (BOC)-β-alanine, BOC-GABA, *l*-histidine methyl ester dihydrochloride, 1-hydroxybenzotriazole monohydrate (HOBt), and 2,2-diphenyl-1-picrylhydrazyl (DPPH) were obtained from Tokyo Chemical Industry (Tokyo, Japan). 1-Methyl-*l*-histidine and [^13^C_3_,^15^N]β-alanine were obtained from Sigma-Aldrich (St. Louis, MO, USA). 3-Methyl-*l*-histidine was obtained from Bachem AG (Torrance, CA, USA). Copper(II) sulfate, 5-hydrate was obtained from Kishida Chemical (Osaka, Japan). Trolox was obtained from Merck (Darmstadt, Germany). All other chemicals and reagents were obtained from common suppliers and were of the highest grade commercially available.

Carnosine and anserine were obtained from Biosynth AG (Staad, Switzerland) and Toronto Research Chemicals (North York, ON, Canada), respectively, and other IDPs (balenine, homocarnosine, and homoanserine) and stable isotope-labeled balenine were synthesized as described below (Section 2.2). 2-Oxo-IDPs and stable isotope-labeled 2-oxo-balenine were prepared as described below (Section 2.3). Other stable isotope-labeled IDPs and 2-oxo-IDPs were prepared as described previously [7].

### 2.2. Synthesis of IDPs

#### 2.2.1. Preparation of Protected Amino Acids

To synthesize methyl ester derivatives of histidine and its methylated analog, 50 mg of histidine or its methylated analog was suspended in methanol (1 mL). Thionyl chloride (200 µL) was added dropwise and the mixture was stirred for 3 h at 50 °C. The solvent was removed in vacuo, resulting in methyl ester derivatives. Next, to prepare BOC derivatives of [^13^C_3_,^15^N]β-alanine, the reaction mixtures of dioxane (250 µL) and 0.5 mM NaOH (250 µL) containing 20 mM [^13^C_3_,^15^N]β-alanine and 60 mM di-*tert*-butyl dicarbonate were stirred for 90 min at room temperature (25 °C). The samples were then dried in vacuo and purified by HPLC (JASCO, Tokyo, Japan) under the following conditions: a Mightysil RP-18 GP column (6.0 × 150 mm; Kanto Chemical, Tokyo, Japan) using a linear gradient of solvent A (water) and solvent B (methanol) (0% B at 0 min; 50% B at 16 min) at a flow rate of 3.0 mL/min. The elution was monitored by UV absorbance at 190 nm. After the concentration of the eluate containing BOC derivatives in vacuo, the concentration of BOC-[^13^C_3_,^15^N]β-alanine was determined by HPLC-UV using the standard curve.

#### 2.2.2. Coupling Reaction

Balenine, homocarnosine, homoanserine, and stable isotope-labeled balenine were synthesized by the carbodiimide coupling reaction [31]. The reaction mixtures (7.5 mL) containing 5 mM methyl ester derivatives of histidine or methylated histidine, 15 mM BOC-β-alanine or BOC-GABA, 10 mM HOBt, 200 mM 2-morpholinoethanesulfonic acid buffer (pH 7.0), and 15 mM EDC were stirred for 2 h on ice, followed by incubation for another 12 h at room temperature (25 °C). After the removal of the solvent in vacuo, the synthesized compound was purified by HPLC using a Scherzo SS-C18 column (10.0 × 150 mm; Imtakt, Kyoto, Japan) and a discontinuous gradient of solvent A (water containing 0.1% formic acid) and solvent B (water containing 50% acetonitrile and 100 mM ammonium formate) (0% B at 0 min; 50% B at 0.1 min; 50% B at 5 min; 90% B at 10 min) at a flow rate of 3.0 mL/min. The elution was monitored by absorbance at 220 nm. The eluate containing the synthesized compound was collected, an equal amount of 1 M NaOH was added to the collected eluate to deprotect methyl ester groups, and the mixture was stirred for 15 min at room temperature (25 °C). Thereafter, the mixture was acidified by adding an equal volume of 6 M HCl and further incubated with an equal volume of 6 M HCl for 90 min at room temperature (25 °C) to deprotect BOC groups. The crude IDP was purified by DOWEX™ 50WX8 cation exchange resin (5 mL; Wako Pure Chemical Corporation, Osaka, Japan) and HPLC using a Scherzo SS-C18 column (10.0 × 150 mm; Imtakt, Kyoto, Japan) under the following conditions: a discontinuous gradient of solvent A (water containing 0.1% formic acid) and solvent B (water containing 50% acetonitrile and 100 mM ammonium formate) (0% B at 0 min; 30% B at 0.1 min; 70% B at 20 min; 100% B at 20.1 min) at a flow rate of 3.0 mL/min. The elution was monitored by absorbance at 220 nm. The purity of synthesized IDPs were analyzed by HPLC-photodiode array (PDA) analysis (Appendix A). To determine the concentrations of synthesized IDPs and stable isotope labeled IDPs, they were hydrolyzed with 6 N HCl for 9 h at 110 °C, and the concentrations of β-alanine or GABA were quantified by HPLC-ESI-MS/MS. Experimental procedures described in this section are shown in Appendix A.

### 2.3. Preparation of 2-Oxo-IDPs

The synthesized or commercially available IDPs were oxidized by the copper/ascorbate oxidation system as described previously [7], with slight modifications. The aqueous mixtures (200 mL) containing 50 mM IDPs and 500 mM ascorbate were adjusted to pH 5.0 with 1 M NH_3_. After the addition of 500 mM CuSO_4_ (2 mL), the mixtures were incubated under oxygen aeration for 90 min at room temperature (25 °C). After the oxidation of IDPs, the reaction mixtures were adjusted to pH 2.0 with 4 M HCl, and then the mixtures were applied to the DOWEX cation exchange column (10 mL) pre-equilibrated with 0.1 M HCl. The flow-through fractions containing 2-oxo-IDPs were collected and applied to another DOWEX cation exchange column (100 mL) pre-equilibrated with 0.1 M HCl. The column was washed with three column volumes of 1 M HCl, and then 2-oxo-IDPs were eluted with three column volumes of 1 M NH_3_. After in vacuo concentration for NH_3_ removal, the samples were diluted with 10 volumes of 80% acetonitrile in water and subjected to silica gel column (Purif-Pack, SI-25, size: 60; Shoko Scientific, Yokohama, Japan). The cartridge was washed with three column volumes of 70% acetonitrile and 2-oxo-IDPs were eluted with three column volumes of 60% acetonitrile. After concentration, the samples were purified by HPLC using a Scherzo SS-C18 column (10.0 × 150 mm; Imtakt) and a discontinuous gradient of solvent A (water containing 0.1% formic acid) and solvent B (water containing 50% acetonitrile and 100 mM ammonium formate) (0% B at 0 min; 20% B at 0.1 min; 20% B at 7 min; 100% B at 7.1 min) at a flow rate of 3.0 mL/min. After the concentration of the eluate containing 2-oxo-IDPs, the samples were diluted with 80% acetonitrile in water and purified by HPLC using an US-Amino column (10.0 × 100 mm; Imtakt) and a discontinuous gradient of solvent A (acetonitrile) and solvent B (water) (0% B at 0 min; 15% B at 0.1 min; 15% B at 5 min; 50% B at 20 min) at a flow rate of 3.0 mL/min. The elution was monitored by absorbance at 250 nm, and the fraction containing 2-oxo-IDPs was concentrated in vacuo. The purity of synthesized 2-oxo-IDPs were analyzed by HPLC-PDA analysis (Appendix A). The chemical structures of the products were characterized by HPLC-ESI-MS/MS. The concentrations of synthesized 2-oxo-IDPs were calculated as described above. Experimental procedures described in this section are shown in Appendix A.

### 2.4. Measurement of Antioxidant Capacity

The DPPH radical scavenging assay was carried out as previously described [7,25], with slight modifications. In brief, in a 96-well plate, 100 μM DPPH was incubated in 12 mM sodium phosphate buffer (pH 7.4) in the presence or absence of 2-oxo-IDPs (final 20–40 μM) or IDPs (final 200–1000 μM) for 20 min at room temperature (25 °C). Absorbance at 517 nm was measured using an Infinite 200 PRO microplate reader (Tecan, Männedorf, Switzerland). Trolox solutions (final 5–30 μM) were used for defining the standard curve. Radical scavenging capacity was evaluated by the inhibition ratio (%), which was calculated by the following formula: Inhibition ratio (%) = (Ac − As)/Ac × 100, where Ac and As indicate the absorbances of the blank control (water) and sample, respectively [25]. The radical scavenging capacity was expressed as Trolox equivalent antioxidant capacity (TEAC), which was calculated using the Trolox standard curve.

### 2.5. Preparation of Meat Extracts

Meat extracts were prepared as described previously [24], with slight modifications. In brief, meat samples (1 g) were shredded with scissors and homogenized using a Heidolph homogenizer (Heidolph, Schwabach, Germany) in a 10-fold volume (*w/v*) of 80% acetonitrile in water containing 50 pmol stable isotope-labeled standards on ice. Next, 50 µL of the homogenate was collected for the analysis of IDP contents and protein quantification and was centrifuged at 12,000× *g* for 20 min at 4 °C. The supernatants were diluted with 0.1% formic acid in water containing 5 pmol stable isotope-labeled standards and subjected to HPLC-ESI-MS/MS. For the analysis of 2-oxo-IDPs, the remaining homogenates were centrifuged at 12,000× *g* for 20 min at 4 °C, and the supernatants were purified using a DOWEX cation exchange column. The eluate was then dried in vacuo, dissolved in 100 µL of 0.1% formic acid in water, and subjected to HPLC-ESI-MS/MS.

### 2.6. Quantitative HPLC-ESI-MS/MS Analysis

IDPs and 2-oxo-IDPs were quantified using a Shimadzu LCMS 8060 triple quadrupole mass spectrometer (Shimadzu, Kyoto, Japan) coupled with the Nexera X2 UHPLC system (Shimadzu) and an Intrada Amino Acid column (3.0 × 100 mm; Imtakt). A discontinuous gradient of solvent A (acetonitrile containing 0.1% formic acid) and solvent B (100 mM ammonium formate) was used as follows: 50% B at 0 min; 50% B at 10 min; 99% B at 11 min, at a flow rate of 0.6 mL/min for the separation of IDPs, and 30% B at 0 min, 35% B at 7 min, 99% B at 7.1 min, at a flow rate of 0.6 mL/min for the separation of 2-oxo-IDPs. The mass spectrometer was operated in the positive mode under the following conditions: nebulizer gas: 3.0 L/min; heating gas: 10 L/min; drying gas: 10 L/min; interface temperature: 350 °C; desolvation line temperature: 250 °C; heat block temperature: 400 °C. LabSolutions LCMS Ver. 5.91 (Shimadzu Scientific, Inc., Columbia, MD, USA) was used for data collection and quantification. Regression equations for the calibration curves were obtained using GraphPad Prism (GraphPad, Inc., La Jolla, CA, USA). The limit of quantification (LOQ) and the limit of detection (LOD) were calculated as the lowest concentration on the calibration curve that was linear and had a signal-to-noise ratio of >10 or >3, respectively.

## 3. Results

### 3.1. Preparation of IDP and 2-Oxo-IDP Standards

Although carnosine and anserine are commercially available from some suppliers, other IDPs (i.e., balenine, homocarnosine, and homoanserine) and stable isotope-labeled IDPs are seldom available. Therefore, we prepared IDPs and stable isotope-labeled IDPs by chemical synthesis using the carbodiimide coupling reaction. The synthesis of IDPs was confirmed by LC-UV detection and HPLC-ESI-MS/MS. Figure 1A shows the result of the MS/MS analysis of balenine. Collision-induced dissociation of balenine revealed major product ions at *m/z* 224.1, 170.1, 124.1, and 109.1. We speculated that the loss of NH_3_ resulted in the formation of the product ion at *m/z* 224.1, and the product ions at *m/z* 170.1, 124.1, and 109.1 originated from a 1-methyl-*l*-histidine moiety. These product ions were identified as shown in Figure 1B. The assignment of these product ions was supported by the observation that the collision-induced dissociation of stable isotope-labeled balenine, containing [^13^C_3_, ^15^N]β-alanine, produced relevant product ions at *m/z* 227.1, 170.1, 124.1, and 109.1 (Appendix A). Similarly, the MS/MS spectra of homocarnosine and homoanserine are shown in Figure 1C,D, and the assignment of product ions was performed (Appendix A). Furthermore, carnosine and anserine were subjected to collision-induced dissociation to compare the MS/MS spectra, and the results are shown in Figure 1E,F. Carnosine and homocarnosine produced common product ions at *m/z* 156.1, 110.1, and 95.1, which can be attributed to the histidine moiety. Similarly, anserine and homoanserine generated common product ions at *m/z* 170.1, 126.1, and 109.1, expected to originate from 3-methyl-*l*-histidine. The characteristic collision-induced dissociation and the assignment of product ion revealed that the IDPs were successfully synthesized. Next, we determined the multiple reaction monitoring (MRM) parameters for the quantification of IDPs based on MS/MS results, and the collision energies were optimized to produce the highest signal intensities of product ions (Table 1).

Next, we synthesized 2-oxo-IDPs by the copper/ascorbate system as described previously [7], with slight modifications. Figure 2A shows the result of the MS/MS analysis of 2-oxo-balenine; major product ions at *m/z* 210.1, 169.1, 123.1, 89.0, and 72.0 were detected. Combined with the results of collision-induced dissociation of stable isotope-labeled 2-oxo-balenine (Appendix A), the losses of CH_2_NH_2_ and OH were speculated to result in the formation of the product ion at *m/z* 210.1, the product ions at *m/z* 169.1 and 123.1 were expected to originate from 2-oxo-1-methyl-*l*-histidine, and the product ions at *m/z* 89.0 and 72.0 were derived from the β-alanine moiety. These product ions were identified as shown in Figure 2B. Similarly, MS/MS spectra of other 2-oxo-IDPs are shown in Figure 2C–F, and the assignment of product ions was performed (Appendix A), revealing the successful synthesis of 2-oxo-IDPs. The MRM parameters for the quantification of 2-oxo-IDPs are shown in Table 2.

Furthermore, to evaluate the chemical property of 2-oxo-IDPs, we performed a DPPH radical scavenging assay. As shown in Figure 3, all of the 2-oxo-IDPs exhibited a greater antioxidant capacity than that of the corresponding precursor; IDPs showed little or no antioxidant capacity despite being used at a higher concentration (i.e., 200–1000 μM) than that of 2-oxo-IDPs (i.e., 20–40 μM). This is the first evidence of the potent antioxidant capacity of 2-oxo-baleinine and 2-oxo-homocarnosine.

### 3.2. Determination of the LC Conditions, LOD, and LOQ for IDPs and 2-Oxo-IDPs

To avoid the co-elution of isomers, we examined various columns and elution condition, and optimized the HPLC condition for the enough separation of dipeptides. Retention times of the peaks of balenine, anserine, homoanserine, carnosine, and homocarnosine were 8.31, 9.29, 9.36, 9.89, and 9.98 min, respectively (Figure 4A). Another peak was detected at the same retention time as that of balenine in the chromatogram of anserine and this peak may be attributed to common precursor ion shared between balenine and anserine (*m/z* 241.3 → 109.1). However, these peaks were adequately separated in the HPLC system and we concluded that these dipeptides were distinguishable on the chromatogram. 2-Oxo-IDPs were also separated under optimized HPLC conditions, and the peaks of 2-oxo-balenine, 2-oxo-anserine, 2-oxo-homoanserine, 2-oxo-carnosine, and 2-oxo-homocarnosine were sufficiently separated and detected at retention times of 3.85, 4.40, 4.57, 4.73, and 4.92 min, respectively (Figure 4B). Although a minor peak was detected in the chromatogram of 2-oxo-anserine at the same retention time as that of 2-oxo-balenine, each peak of 2-oxo-IDPs was sufficiently separated.

To determine the LOD and LOQ of IDPs, standard calibration curves were generated in the concentration range of 1–1000 fmol. Appendix A shows the regression equation for the calibration curve of IDPs and revealed that IDPs exhibited good linearity (*r*^2^ = 0.999). Similarly, the standard calibration curves of 2-oxo-IDPs were constructed in same concentration range and showed good linearity (*r*^2^ = 0.999) (Appendix A). An analysis of the signal-to-noise ratio revealed that the LOQ values of IDPs and 2-oxo-IDPs were in range of 240.1–2561 ng/mL by this method (Table 3), corresponding to in range of 10–100 fmol. Furthermore, the lowest LOD was 3 fmol (72.03 ng/mL) for IDPs and 2-oxo-IDPs.

### 3.3. Quantitative Identification of IDPs and 2-Oxo-IDPs in Meat Samples

We analyzed IDPs and 2-oxo-IDPs in several meat samples by HPLC-ESI-MS/MS. Representative MS/MS chromatograms of IDPs are shown in Figure 5A. Endogenous IDPs were co-eluted at the same retention time as those of the spiked stable isotope-labeled standards. Representative MS/MS chromatograms of 2-oxo-IDPs showed that five endogenous 2-oxo-IDPs were detected simultaneously with each internal standard (Figure 5B).

The quantitative analyses of IDPs and 2-oxo-IDPs are summarized in Table 4. All IDPs were detected in all samples in this study, except for homoanserine in whale meat. Carnosine was a predominant IDP in beef and pork, whereas anserine and balenine were the major IDPs in chicken and whale meat, respectively. Homocarnosine and homoanserine were also detected, although the levels of these IDPs were considerably lower than those of major IDPs. The total concentrations of IDPs in beef, pork, chicken, and whale meat were 250, 340, 300, and 690 nmol/mg protein, respectively.

2-Oxo-carnosine, 2-oxo-anserine, and 2-oxo-balenine were detected in all samples analyzed in this study. 2-Oxo-homocarnosine and 2-oxo-homoanserine were also detected in beef, whereas these dipeptides were detectable but unquantifiable or undetectable in other meat samples. The total contents of 2-oxo-IDPs were 280, 150, 250, and 100 pmol/mg protein in beef, pork, chicken, and whale meat, accounting for 0.11%, 0.044%, 0.082%, and 0.015% of the total IDPs.

## 4. Discussion

In the current study, we synthesized IDPs for the preparation of 2-oxo-IDPs and internal standards for accurate MS analyses. Although carnosine and anserine are commercially available from suppliers, the availability of other IDPs (e.g., balenine, homocarnosine, and homoanserine) and stable isotope-labeled IDPs is limited. Here, we synthesized several types of IDPs, including stable isotope-labeled balenine, by the carbodiimide coupling reaction. By HPLC-ESI-MS/MS, anserine, balenine, and homocarnosine were detected as precursor ions at *m/z* 241; however, MS/MS spectra of the fragment ions from these IDPs differed slightly. Turecek et al. reported that the precursor ion of 1-methyl-*l*-histidine (component of balenine), which appeared at *m/z* 170, underwent the dominant loss of H_2_O + CO to form a fragment ion at *m/z* 124; in contrast, 3-methyl-*l*-histidine (component of anserine) showed a loss of CO_2_ to form a fragment ion at *m/z* 126 as important primary dissociations [32]. They also demonstrated that these differences in dissociation were related to a difference in the methylation site in the imidazole ring. Uenoyama et al. reported that MS/MS analyses of balenine produced the highest ion peak at *m/z* 124, whereas those of anserine produced the highest ion peak at *m/z* 109 [33]. These findings are consistent with our observations, indicating that the IDPs were successfully synthesized. We also prepared 2-oxo-IDPs and stable isotope-labeled 2-oxo-balenine by the copper/ascorbate oxidation system [7], using synthesized IDPs and stable isotope-labeled IDPs, respectively, as starting materials. The results of MS/MS analyses were consistent with previous results [7,23], suggesting the successful synthesis of 2-oxo-IDPs. Consequently, we successfully prepared five IDPs, five 2-oxo-IDPs, and their stable isotope-labeled compounds. The use of these compounds as standards enabled their quantitative detection by HPLC-ESI-MS/MS.

HPLC-ESI-MS/MS has been used for the detection and quantitation of IDPs [30]. Macia et al. developed a LC-MS method for the rapid, sensitive, and selective determination of carnosine and anserine in samples containing complex matrices [34]. However, matrix effects, such as ion suppression, must be considered during HPLC-ESI-MS/MS-based detection to achieve reliable analytical data [35]. To correct for this effect, an isotope dilution method has been used for the exact quantitative MS analysis of carnosine [31,36], anserine [31,37], homocarnosine [38], and homoanserine [23]. However, no study has used isotope-labeled balenine. The application of HPLC-ESI-MS/MS coupled with a stable isotope dilution method for 2-oxo-IDPs has only been demonstrated by our previous reports for the detection of 2-oxo-carnosine [7,24,25], 2-oxo-anserine [7,24,25], 2-oxo-homocarnosine [7], and 2-oxo-homoanserine [23]. However, no study has demonstrated about 2-oxo-balenine. In this study, we developed quantitative HPLC-ESI-MS/MS methods using isotope-labeled balenine and 2-oxo-balenine. To the best of our knowledge, this is the first demonstration of the quantitative analysis of balenine and 2-oxo-balenine using HPLC-ESI-MS/MS coupled with a stable isotope dilution method. Together with our previous reports, it is now possible to detect IDPs and 2-oxo-IDPs specifically and quantitatively by using these two different HPLC programs followed by MS/MS analysis, respectively. Moreover, we demonstrated the highly sensitive and selective detection of IDPs and 2-oxo-IDPs in MRM mode using HPLC-ESI-MS/MS and achieved detection at the level of 3 fmol. Jozanović et al. summarized recent analytical methods for the detection of carnosine, reporting LOD values in the range of 4.5–7000 fmol [39]. The LOD of our current method is comparable to or lower than those of other analytical techniques. Therefore, the newly established method using HPLC-ESI-MSMS combined with stable isotope dilution enabled the highly sensitive and exact detection of five 2-oxo-IDPs.

We determined the contents of five IDPs and five 2-oxo-IDPs in beef, pork, chicken, and whale meat samples by HPLC-ESI-MSMS coupled with a stable isotope dilution method. As shown in Table 4, we successfully detected five IDPs from all meat samples used in this study, and the contents and proportion of IDPs in various sample types were approximately consistent with those of previous reports [6,33,40]. Various studies have evaluated IDP contents in meat; however, many of them focused only on the content of predominant IDPs. In fact, only a few studies have reported the content of homocarnosine in meat [41,42], and studies of the homoanserine content in meat are lacking. We successfully detected homoanserine in meat, including beef, pork, and chicken, for the first time using the newly developed method. These results support the use of our quantitative method for IDP detection and the evaluation of minor constituents in meat. Moreover, analytical methods based on stable isotope dilution provide reliable data on the absolute amounts of several IDPs, providing a basis for comparative analyses of IDP compositions in meat.

We successfully identified several 2-oxo-IDPs in meat in a single measurement. Notably, this is the first evidence for the endogenous production of 2-oxo-balenine, which was detected in not only whale but also in other meat types. The total amount of 2-oxo-IDPs was 100–280 pmol/mg protein and the percentage of 2-oxo-IDPs to IDPs was 0.015–0.11%. We have previously reported 2-oxo-IDP contents in meat of 48–340 pmol/mg protein, accounting for 0.027–0.26% of the total IDP contents [24], in concordance with the results of the current study. We also showed that the predominant types of 2-oxo-IDPs were consistent with the predominant IDPs, indicating that the amounts of 2-oxo-IDPs were correlated with those of IDPs in meat. This observation is in line with previous results [7,23]. Thus, the highly sensitive and specific detection method developed here provides precise insight into the content and distribution of 2-oxo-IDPs.

Previous studies have demonstrated that 2-oxo-IDPs (i.e., 2-oxo-carnosine, 2-oxo-anserine, and 2-oxo-homoanserine) exhibit a greater antioxidant capacity, compared to the corresponding IDPs [7,23,25]. Further, it has been reported that 2-oxo-carnosine shows significant cytoprotective effects on rotenone-induced neuronal cell death, while there are no significant effects by the treatment with the precursor carnosine [7]. In addition, in the current study, the DPPH radical scavenging assay revealed that all 2-oxo-IDPs synthesized herein, including 2-oxo-balenine and 2-oxo-homocarnosine, exhibited a potent antioxidant capacity, while little activities by the precursor IDPs were observed (Figure 3). These results are consistent with previous results of the evaluation of the antioxidant capacity of 2-oxo-IDPs and IDPs [18,19,20,21,22,25], suggesting that 2-oxo-IDPs can function as potent antioxidants in vivo, and the conversion of IDPs to 2-oxo-IDPs may be an important process in the physiological functions of IDPs. To evaluate the stability of 2-oxo-IDPs under physiological conditions, pharmacokinetic analyses have to be performed. By further detailed studies to evaluate the metabolism and pharmacological characters of 2-oxo-IDPs using the detection method for 2-oxo-IDPs developed in this study, the contents of not only IDPs but also 2-oxo-IDPs in foods such as meat and its derivatives may be a novel evaluation criterion for the effectiveness, such as the quality, health benefits, and preservation.

## 5. Conclusions

We established a highly sensitive, selective, and absolute quantitative analytical method for detecting five IDPs and five 2-oxo-IDPs, novel oxidized derivatives of IDPs, using HPLC-ESI-MS/MS coupled with a stable isotope dilution method utilizing stable isotope-labeled IDP and 2-oxo-IDP standards. Furthermore, we successfully detected not only IDPs but also 2-oxo-IDPs in the meats of several vertebrates and revealed that the 2-oxo-IDP content is generally several orders of magnitude lower than corresponding IDP contents. Moreover, we demonstrated, for the first time, that 2-oxo-balenine is endogenously produced in meats. The highly sensitive detection technique enables the detection of trace levels of 2-oxo-IDPs in biological samples and could contribute to our understanding of the biological relevance of 2-oxo-IDPs, which are highly functional in small amounts.

## Figures and Tables

**Figure 1 antioxidants-11-02401-f001:**
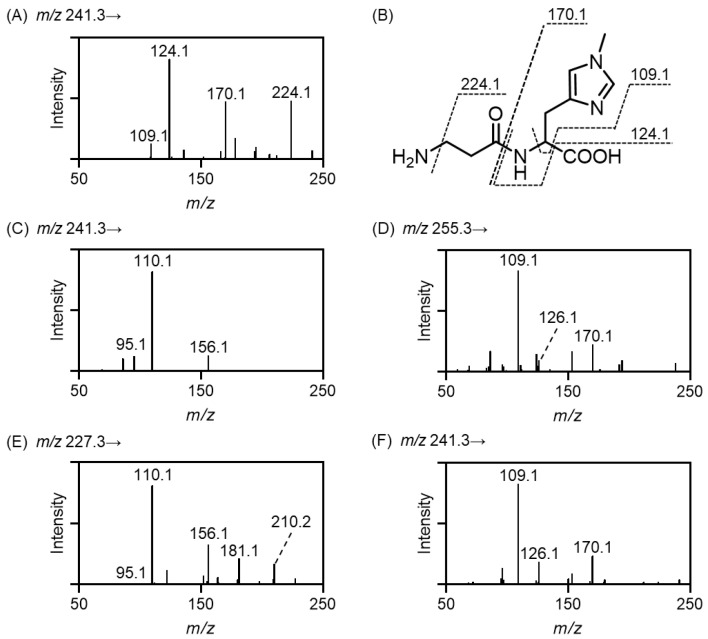
MS/MS spectra of IDPs. IDPs were synthesized by the carbodiimide coupling reaction and subjected to HPLC-ESI-MS/MS. MS/MS spectra of balenine (**A**), homocarnosine (**C**), homoanserine (**D**), carnosine (**E**), and anserine (**F**) are shown, respectively. Product ions of balenine were assigned as shown in (**B**).

**Figure 2 antioxidants-11-02401-f002:**
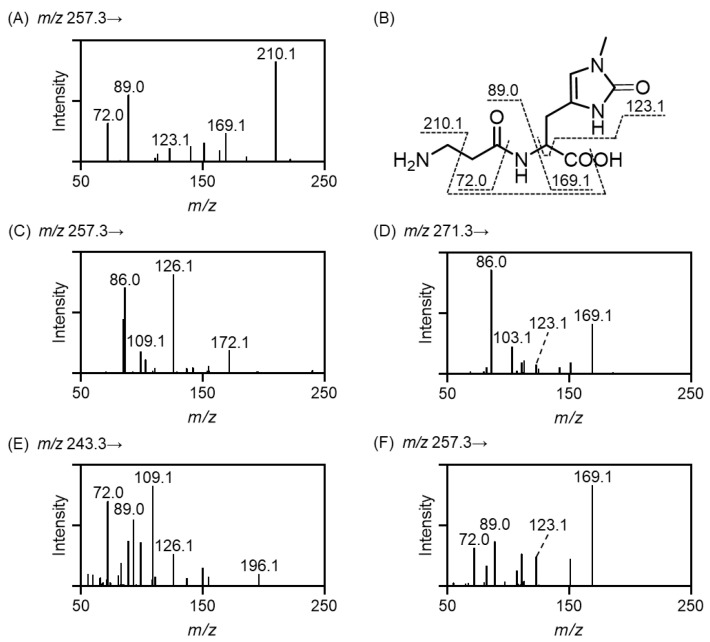
MS/MS spectra of 2-oxo-IDPs. 2-Oxo-IDPs were prepared by the copper/ascorbate oxidation system and subjected to HPLC-ESI-MS/MS. MS/MS spectra of 2-oxo-balenine (**A**), 2-oxo-homocarnosine (**C**), 2-oxo-homoanserine (**D**), 2-oxo-carnosine (**E**), and 2-oxo-anserine (**F**) are shown. The assignment of the product ions of 2-oxo-balenine were conducted as shown in (**B**).

**Figure 3 antioxidants-11-02401-f003:**
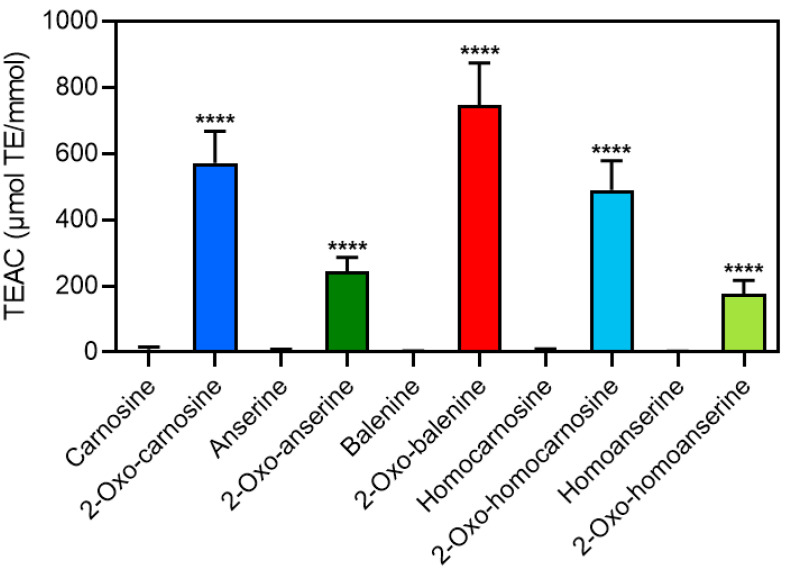
Antioxidant capacity of 2-oxo-IDPs and IDPs. Antioxidant capacity of 2-oxo-IDPs and IDPs were evaluated by DPPH radical scavenging assay. Data are expressed as Trolox equivalent antioxidant capacity (TEAC): μmol Trolox equivalent (TE) per mmol samples. Data are presented as means ± standard deviation (SD) (*n* > 6). ********
*p* < 0.0001 versus the corresponding IDPs, compared using unpaired Student’s *t* test.

**Figure 4 antioxidants-11-02401-f004:**
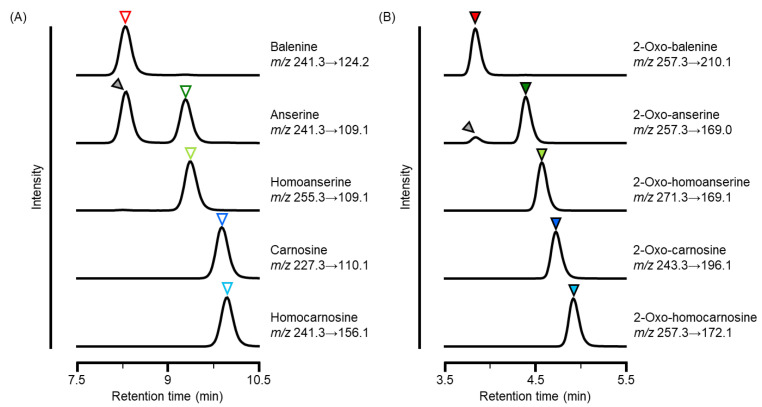
Separation of several IDPs and 2-oxo-IDPs using HPLC-ESI-MS/MS. A standard mixture of IDPs and 2-oxo-IDPs (1 µM each) was subjected to HPLC-ESI-MS/MS using an Intrada Amino Acid column. (**A**) Representative MS/MS chromatograms of IDPs. Open triangles indicate each IDP, and the gray triangle indicates balenine detected in MRM for anserine. (**B**) Representative MS/MS chromatograms of 2-oxo-IDPs. Filled triangles indicate each 2-oxo-IDP, and gray triangle indicates 2-oxo-balenine detected in MRM for 2-oxo-anserine.

**Figure 5 antioxidants-11-02401-f005:**
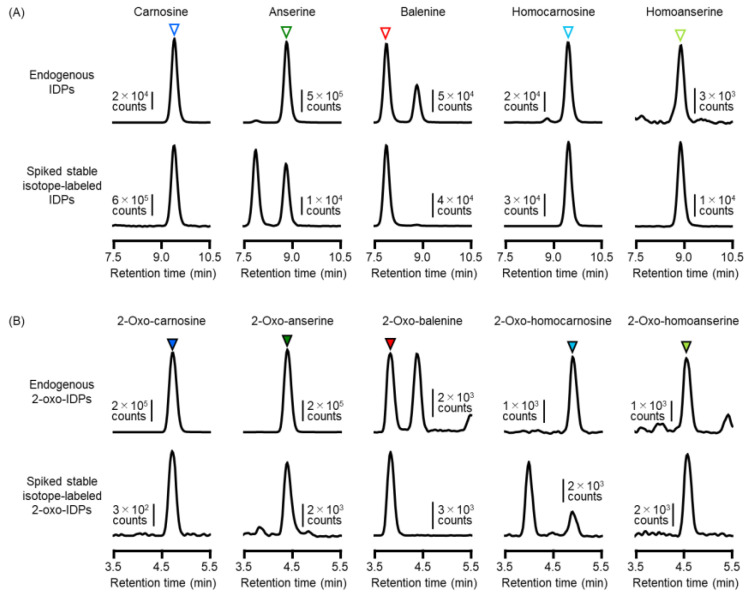
HPLC-ESI-MS/MS analysis of IDPs and 2-oxo-IDPs in beef. Meat samples were homogenized in 80% acetonitrile/water containing stable isotope-labeled standards, and extracted IDPs and 2-oxo-IDPs were subjected to HPLC-ESI-MS/MS analysis. (**A**) Representative chromatograms of endogenous IDPs (upper trace), and spiked isotope-labeled IDPs (lower trace) are shown. Open triangles indicate each IDP. (**B**) Representative chromatograms of endogenous 2-oxo-IDPs (upper trace), and spiked isotope-labeled 2-oxo-IDPs (lower trace) are shown. Filled triangles indicate each 2-oxo-IDP.

**Table 1 antioxidants-11-02401-t001:** MRM parameters for IDPs and stable isotope-labeled IDPs.

Analyte	Precursor Ion (*m/z*)	Product Ion (*m/z*)	Collision Energy (V)
Carnosine	227.3	110.1	25
Carnosine *	231.0	110.1	25
Anserine	241.3	109.1	25
Anserine *	245.0	109.1	25
Balenine	241.3	124.2	25
Balenine *	245.3	124.2	25
Homocarnosine	241.3	156.1	10
Homocarnosine *	244.1	159.1	10
Homoanserine	255.3	109.1	25
Homoanserine *	261.1	109.1	25

Collision energies were optimized to produce the highest signal intensities of product ions. Asterisks indicate isotope-labeled IDPs.

**Table 2 antioxidants-11-02401-t002:** MRM parameters for 2-oxo-IDPs and stable isotope-labeled 2-oxo-IDPs.

Analyte	Precursor Ion (*m/z*)	Product Ion (*m/z*)	Collision Energy (V)
2-Oxo-carnosine	243.3	196.1	10
2-Oxo-carnosine *	247.1	198.0	10
2-Oxo-anserine	257.3	169.0	15
2-Oxo-anserine *	261.1	169.0	15
2-Oxo-balenine	257.3	210.1	15
2-Oxo-balenine *	261.3	212.1	15
2-Oxo-homocarnosine	257.3	172.1	15
2-Oxo-homocarnosine *	260.1	175.1	15
2-Oxo-homoanserine	271.3	169.1	15
2-Oxo-homoanserine *	277.2	169.1	15

Collision energies were optimized to produce the highest signal intensities of product ions. Asterisks indicate isotope-labeled 2-oxo-IDPs.

**Table 3 antioxidants-11-02401-t003:** LOD and LOQ of IDPs and 2-oxo-IDPs.

	Carnosine	Anserine	Balenine	Homocarnosine	Homoanserine
LOD (ng/mL)	226.1	240.1	72.03	72.03	254.1
LOQ (ng/mL)	678.3	720.3	240.1	240.1	762.3
	**2-Oxo-Carnosine**	**2-Oxo-Anserine**	**2-Oxo-Balenine**	**2-Oxo-Homocarnosine**	**2-Oxo-Homoanserine**
LOD (ng/mL)	72.63	256.1	76.83	256.1	270.1
LOQ (ng/mL)	242.1	768.3	256.1	2561	810.3

LOD and LOQ were calculated as the lowest concentration on the calibration curve that was linear and had a signal-to-noise ratio of >3 or >10, respectively.

**Table 4 antioxidants-11-02401-t004:** Quantitative HPLC-ESI-MS/MS analysis of IDPs and 2-oxo-IDPs in meat samples.

	IDPs (nmol/mg Protein)				
Meat	Carnosine	Anserine	Balenine	Homocarnosine	Homoanserine
Beef	220 ± 19	31 ± 2.6	2.1 ± 1.9	0.48 ± 0.10	0.091 ± 0.014
Pork	320 ± 23	5.2 ± 0.48	14 ± 1.1	0.036 ± 0.0040	0.0029 ± 0.00018
Chicken	110 ± 6.5	190 ± 5.3	5.8 ± 0.82	0.031 ± 0.0056	0.021 ± 0.00094
Whale	19 ± 0.83	4.4 ± 0.39	670 ± 23	0.0090 ± 0.0013	N.D.
	**2-Oxo-IDPs (pmol/mg Protein)**				
**Meat**	**2-Oxo-Carnosine**	**2-Oxo-Anserine**	**2-Oxo-Balenine**	**2-Oxo-Homocarnosine**	**2-Oxo-Homoanserine**
Beef	230 ± 120	51 ± 7.9	0.44 ± 0.38	0.46 ± 0.19	0.22 ± 0.042
Pork	140 ± 47	5.6 ± 0.15	1.9 ± 0.45	N.Q.	N.D.
Chicken	66 ± 11	180 ± 27	0.73 ± 0.080	N.D.	N.Q.
Whale	18 ± 6.2	2.2 ± 0.35	81 ± 28	N.D.	N.D.

Data are presented as means ± SD (*n* = 3). N.Q.; not quantified. N.D.; not detected.

## Data Availability

Data are contained within the article and Appendix A.

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
