# Peer review of "Quantitative Determination of 2-Oxo-Imidazole-Containing Dipeptides by High-Performance Liquid Chromatography/Tandem Mass Spectrometry"

_antioxidants, 2022, doi:10.3390/antiox11122401_

Round 1

Reviewer 1 Report

The work presented here deals with a potentially interesting topic, is exhaustive, and is scientifically well-developed. I do not have too many aspects or critical points to comment on in the paper itself. My biggest concern is the nature of the study for a journal like Antioxidants. In this sense, although the great antioxidant capacity of the analytes is repeatedly mentioned, no result, assay, survey, etc. supporting this issue is provided. Anyway, if the Scientific Editor thinks that the paper meets the aims and scope of antioxidants, my only comment would be that the authors should try to emphasize the antioxidant character of the compounds with experimental or bibliographic data in this sense.

On the other hand, some additional aspects to be addressed are as follows:

Section 2.2. Please, provide details of the fractions collected and their further treatment.

Line 250. Briefly comment on the studies performed to optimize the separation.

LODs and LOQs in terms of amount (here expressed as fmol) may be usual in some instances but such data in terms of concentration will be useful as well. I suggest providing complementary data in mass/volume (e.g., ng/mL, pg/mL). Then, we will understand better the overall detection capacity of the methods.

Table 4. Express the results with a suitable number of significant figures. For instance, 219.21 ± 11.06 should be 220 ± 10, and so on. Indicate that  ± xx corresponds to the standard deviation (n= 3?)

Line 369. Table 5 does not exist.

Author Response

Reviewer#1

Comment-1: The work presented here deals with a potentially interesting topic, is exhaustive, and is scientifically well-developed. I do not have too many aspects or critical points to comment on in the paper itself. My biggest concern is the nature of the study for a journal like Antioxidants. In this sense, although the great antioxidant capacity of the analytes is repeatedly mentioned, no result, assay, survey, etc. supporting this issue is provided. Anyway, if the Scientific Editor thinks that the paper meets the aims and scope of antioxidants, my only comment would be that the authors should try to emphasize the antioxidant character of the compounds with experimental or bibliographic data in this sense.

Response: We appreciate these insightful and positive comments.  According to the reviewer’s comment, we evaluated the antioxidant capacity of 2-oxo-IDPs and IDPs by in vitro assay.  The result demonstrates that all 2-oxo-IDPs analyzed in this study exhibited a greater antioxidant capacity than did the corresponding IDPs.  This is the first evidence of the potent antioxidant capacity of 2-oxo-balenine and 2-oxo-homocarnosine.  This result is in line with previous reports and emphasize the potentials of 2-oxo-IDPs as potent antioxidants in vivo.  Therefore, we believe that the contents of not only IDPs but also 2-oxo-IDPs in foods such as meat and its derivatives may be a novel evaluation criterion for the effectiveness, such as the quality, health benefits, and preservation.  We have mentioned these points briefly in the Methods (P. 2, L. 92–93; P. 4, L. 179–190 in the revised manuscript), Results (P. 7, L. 277–P. 8, L. 289 in the revised manuscript), and Discussion (P. 12, L. 428–439 in the revised manuscript), and added a new figure (Figure 3) to the revised manuscript.

Comment-2: Section 2.2. Please, provide details of the fractions collected and their further treatment.

Response: We apologize for our insufficient explanation.  We have added the explanations for the details of the fractions collected and further treatment (P. 3, L. 119–120 in the revised manuscript).

Comment-3: Line 250. Briefly comment on the studies performed to optimize the separation.

Response: Thank you for this comment.  We added a sentence about the optimization of analytes separation (P. 8, L. 291–292 in the revised manuscript).

Comment-4: LODs and LOQs in terms of amount (here expressed as fmol) may be usual in some instances but such data in terms of concentration will be useful as well. I suggest providing complementary data in mass/volume (e.g., ng/mL, pg/mL). Then, we will understand better the overall detection capacity of the methods.

Response: Thank you for this valuable comment.  According to the reviewer’s suggestion, we have modified the sentences (P. 9, L. 317–320 in the revised manuscript) and Table 3.

Comment-5: Table 4. Express the results with a suitable number of significant figures. For instance, 219.21 ± 11.06 should be 220 ± 10, and so on. Indicate that ± xx corresponds to the standard deviation (n= 3?).

Response: According to the reviewer’s suggestion, we have modified the table (Table 4) and added an explanation about the data in the caption (P. 11, L. 347), Results (P. 10, L. 344–345; P. 11, L. 352–354), and Discussion (P. 12, L. 419–421).

Comment-6: Line 369. Table 5 does not exist.

Response: Thank you for pointing this out. We have revised manuscript (P. 12, L. 405). 

Reviewer 2 Report

The study is well written and the results are clearly presented. However, manuscript is typical analytic work thus in my opinion it doesn't fit to Antioxidants. Authors should better highlighted the role of the analysed compounds as an antioxidant  in Discussion.

As experimental procedures are complex, scheme of experimental design could be added in sections 2.2.2, 2.3

Lines 182-183: it seems that two different elution programs were used to separate 2-oxo-IDPs and IDPs; therefore analyses were not carried out in one chromatographic run. But in line 358 Authors stated “it is now possible to quantitatively analyze five IDPs and five 2-oxo-IDPs in one measurement, respectively.” – Could you explain?

Figure 1 and 2: “HPLC-ESI-MS/MS analysis of 2-oxo-IDPs”-  Precisely the figures show m/s spectra

Figure 4. Regression equation for the calibration curves – It is not necessary in main body (it can be moved to Supplementary material)

Author Response

Reviewer#2

Comment-1: The study is well written and the results are clearly presented. However, manuscript is typical analytic work thus in my opinion it doesn't fit to Antioxidants. Authors should better highlighted the role of the analysed compounds as an antioxidant  in Discussion.

Response: We are very much grateful for the reviewer’s helpful and positive comments.  To highlight the importance of 2-oxo-IDPs as potent antioxidants and to fit this study to the journal of Antioxidants, we performed an additional experiment to evaluate the antioxidant capacity of 2-oxo-IDPs by DPPH radical scavenging assay.  The result demonstrates that all 2-oxo-IDPs analyzed in this study exhibited a greater antioxidant capacity than did the corresponding IDPs.  This is the first evidence of the potent antioxidant capacity of 2-oxo-balenine and 2-oxo-homocarnosine.  This result is in line with previous reports and emphasize the potentials of 2-oxo-IDPs as potent antioxidants in vivo.  Therefore, we believe that the contents of not only IDPs but also 2-oxo-IDPs in foods such as meat and its derivatives may be a novel evaluation criterion for the effectiveness, such as the quality, health benefits, and preservation.  We have mentioned these points briefly in the Methods (P. 2, L. 92–93; P. 4, L. 179–190 in the revised manuscript), Results (P. 7, L. 277–P. 8, L. 289 in the revised manuscript), and Discussion (P. 12, L. 428–439 in the revised manuscript), and added a new figure (Figure 3) to the revised manuscript.

Comment-2: As experimental procedures are complex, scheme of experimental design could be added in sections 2.2.2, 2.3

Response: According to the reviewer’s suggestion, we have added new figures of schematic diagram of experiments described in the sections 2.2.2 and 2.3 in the revised Supplementary Materials (Supplementary Figures S2 and S3), and mentioned the explanations to Methods (P. 3, L. 147–148; P. 4, L. 177–178 in the revised manuscript).

Comments-3: Lines 182-183: it seems that two different elution programs were used to separate 2-oxo-IDPs and IDPs; therefore analyses were not carried out in one chromatographic run. But in line 358 Authors stated “it is now possible to quantitatively analyze five IDPs and five 2-oxo-IDPs in one measurement, respectively.” – Could you explain?

Response: We apologize for our insufficient descriptions.  To avoid confusion, we modified the sentence as follows: it is now possible to detect IDPs and 2-oxo-IDPs specifically and quantitatively by using these two different HPLC programs followed by MS/MS analysis, respectively (P. 11, L. 394–396 in the revised manuscript).

Comments-4: Figure 1 and 2: “HPLC-ESI-MS/MS analysis of 2-oxo-IDPs”-  Precisely the figures show m/s spectra.

Response: We modified the title of the figures: “MS/MS spectra of IDPs” (for Figure 1, P. 6, L. 248 in the revised manuscript) and “MS/MS spectra of 2-oxo-IDPs” (for Figure 2, P. 7, L. 269 in the revised manuscript).

Comments-5: Figure 4. Regression equation for the calibration curves – It is not necessary in main body (it can be moved to Supplementary material).

Response: According to the reviewer’s comment, we have moved the figure (Figure 4 in the original manuscript) to the revised Supplementary Materials (Supplementary Figure S6).

Reviewer 3 Report

Dear Authors

I found this work interesting and suitable for publication in this SI of Antioxidants. The text is correctly organized and the style is coherent with he journal guidelines. I have the following points to address before publication:

1. Please check the language for typos and grammar.

2. Authors should provide analytical data for the synthesized peptide, e.g. HPLC traces for purity at 3 diverse wavelights, NMR characterization 1H and 13C.

3. Since this is a synthetic peptide, I suggest the authir to mention the strategy applied in light of the molecule's design. At this regard some information and useful background can be found in the following literature: "Design, Synthesis and Biological Evaluation of Two Opioid Agonist and Cav 2.2 Blocker Multitarget Ligands"; "Antinociceptive profile of potent opioid peptide AM94, a fluorinated analogue of biphalin with non-hydrazine linker".

4. Stability checking of the peptide in human fluid should be considered.

Author Response

Reviewer#3

Comment-1: I found this work interesting and suitable for publication in this SI of Antioxidants. The text is correctly organized and the style is coherent with he journal guidelines. I have the following points to address before publication:

Response: We are very much grateful for the reviewer’s helpful and positive comments.  We have responded to each comment below in a point-by-point manner.  We hope that we have satisfactorily addressed all comments and that manuscript has been improved.

Comment-2: Please check the language for typos and grammar.

Response: The manuscript has been checked by an English language editing service before re-submitting of the revised manuscript.

Comment-3: Authors should provide analytical data for the synthesized peptide, e.g. HPLC traces for purity at 3 diverse wavelights, NMR characterization 1H and 13C.

Response: According to the reviewer’s comment, we added a new figure of the analysis of the synthesized IDPs and 2-oxo-IDPs by HPLC-photodiode array (Supplementary Figure S1 in the revised Supplementary materials), and mentioned the explanations to Methods (P. 3, L. 143–144; P. 4, L. 174–175 in the revised manuscript).

Comment-4: Since this is a synthetic peptide, I suggest the authir to mention the strategy applied in light of the molecule's design. At this regard some information and useful background can be found in the following literature: "Design, Synthesis and Biological Evaluation of Two Opioid Agonist and Cav 2.2 Blocker Multitarget Ligands"; "Antinociceptive profile of potent opioid peptide AM94, a fluorinated analogue of biphalin with non-hydrazine linker".

Response: We are thankful for the valuable comments.  The target IDPs are well-characterized endogenous substances and known for a broad spectrum of bioactivities, however, the commercially availability of several IDPs (i.e., balenine, homocarnosine, and homoanserine) and stable isotope-labeled IDPs is limited.  In addition, it has revealed that 2-oxo-IDPs, such as 2-oxo-carnosine, 2-oxo-anserine, and 2-oxo-homoanserine, are endogenously produced in vivo, and the oxidized IDPs derivatives exhibit a greater antioxidant capacity than the precursor IDPs do.  The synthesis of these peptides is for the preparation of standards for the measurement of endogenous peptides, not for molecular design.  Although molecular design of IDP derivatives with new functions is attractive, this study is limited to the synthesis of standard IDPs and 2-oxo-IDPs, including stable isotope labeled-peptides, and the establishment of quantitative determination of IDPs.  We have mentioned these points briefly in the Introduction (P. 2, L. 62–73 in the revised manuscript) and the Results (P. 5, L. 222–224 in the revised manuscript).

Comment-5: Stability checking of the peptide in human fluid should be considered.

Response: Thank you for the insightful comment.  This point is of great interest to us and further studies to evaluate the pharmacological characteristics of 2-oxo-IDPs remain to be done.  However, this issue is quite far from the subject of the current study, and we will address this issue in a future study.  We added a brief explanation of the importance of pharmacokinetic analyses of 2-oxo-IDPs in the Discussion (P. 12, L. 439–444).

Round 2

Reviewer 1 Report

The paper has been improved substantially. All my concerns have been addressed properly.